# Subclinical Arteriosclerosis is Associated with Common Vascular Risk Factors in Long-Term Survivors of Testicular Cancer

**DOI:** 10.3390/jcm9040971

**Published:** 2020-03-31

**Authors:** Javier Espíldora-Hernández, Tania Díaz-Antonio, Javier Baena-Espinar, Inmaculada Alonso-Calderón, José Rioja, Emilio Alba-Conejo, Pedro Valdivielso, Miguel-Ángel Sánchez-Chaparro

**Affiliations:** 1Internal Medicine Unit, Hospital Virgen de la Victoria, 29010 Málaga, Spain; javierespildora@gmail.com (J.E.-H.); masanchezchaparro@gmail.com (M.-Á.S.-C.); 2Radiology Unit, Hospital Virgen de la Victoria, 29010 Málaga, Spain; tdiazant@gmail.com; 3Oncology Unit, Hospital Virgen de la Victoria, 29010 Málaga, Spain; javibaes@gmail.com (J.B.-E.); ealbac@uma.es (E.A.-C.); 4Preventive Center for Civil Servant, Consejería de Empleo, 29010 Málaga, Spain; minmaculada.alonso@juntadeandalucia.es; 5Lipid and Arteriosclerosis Laboratory, Department of Medicine and Dermatology, and Biomedical Institute for Research (IBIMA), Universidad de Málaga, 29010 Málaga, Spain; jose.rioja@uma.es

**Keywords:** testicular cancer, subclinical arteriosclerosis, liver steatosis, arterial ultrasound, vascular calcification

## Abstract

Cardiovascular disease risk is increased in survivors of testicular cancer because of exposure to treatment (chemotherapy and radiotherapy), as well as modification in lifestyle. Our aim was to assess the presence of subclinical arteriosclerosis in survivors of testicular cancer in comparison with a control group. This was a cross-sectional, observational, case–control study including 50 survivors of Germ Cell Tumor (GCT) (14 years of follow-up) and 53 age-matched controls with no cancer. We registered clinical data, cardiovascular risk factors, physical and Mediterranean questionnaires, intima-media thickness and plaque at carotid and femoral arteries by ultrasound, calcium score at the abdominal aorta, and liver steatosis by computed tomography, and applied analytical tests to quantify metabolic risk factors and inflammation markers. Patients showed a trend toward greater intima-media thickness (IMT) and plaques than controls, as well as a higher calcium score in the abdominal aorta. Remarkably, patients had higher waist circumference, insulin resistance (HOMA-IR), and liver steatosis, but lower physical activity and high-density lipoprotein (HDL) cholesterol than controls (all *p* < 0.05). In multivariate analyses, only common vascular risk factors were associated with subclinical arteriosclerosis. As a conclusion, in our study, a higher rate of subclinical arteriosclerosis in testicular cancer survivors was associated with classical metabolic risk factors and lifestyle, but not with exposure to chemotherapy.

## 1. Introduction

Cardiovascular disease is a major concern in long-term survivors of cancer [1,2,3], especially if they are young [4]. This is the case for testicular cancer, the most common malignancy among men 14 to 44 years of age [5]. In testicular cancer subjects, cardiovascular disease (CVD) risk was found to be almost double that of the general population in four large epidemiological studies, and was associated with the use of chemotherapy and mediastinal radiotherapy [6,7,8,9]. In fact, a recent consensus conference reinforced the role of preventive measures to avoid late cardiovascular disease after treatment of testicular germ cell tumor (GCT) [10]. Moreover, more than half of long-term GCT survivors have a sedentary lifestyle [11], up to 39% continue to smoke, and metabolic syndrome affects up to 16% [12].

Atherosclerosis is a progressive disease characterized by the accumulation of lipids and fibrous elements in large arteries, which finally evolves towards arterial calcification [13]; it remains asymptomatic for many years until the plaques break, and the disease is complicated by thrombosis or vascular occlusion is done by significant stenosis [14]. Subclinical arteriosclerosis (SA) can be assessed by the presence of plaques in carotid and femoral arteries as well as calcification in the abdominal aorta. Both are independent predictors of ischemic vascular events and death [15,16].

The aim of this study was to analyze subclinical arteriosclerosis among testicular cancer survivors in comparison with an age-matched control group.

## 2. Patients and Methods

### 2.1. Study Design

This was a cross-sectional, observational case–control study. Cases were obtained from the Oncology Department at the University Hospital Virgen de la Victoria (Málaga, Spain). Age-matched controls were selected from Centro Provincial de Prevención de Riesgos Laborales, Consejería de Empleo, Málaga, Spain.

Inclusion criteria for cases were male adults (≥18 years), with a history of GCT treated with chemotherapy. Among 126 subjects regularly seen in our outpatient clinic, 98 could be contacted by phone and 61 accepted initially. Finally, only 50 patients signed the informed consent and completed the study. The inclusion criteria for the controls were healthy men with same demographic characteristics. Children and those with previous cardiovascular events were excluded.

We recorded demographic data, age at GCT diagnosis, tumor histology, applied chemotherapy, cardiovascular risk factors, cardiovascular risk according to the Systematic *Coronary Risk* Evaluation (SCORE) system [17,18], adherence to a Mediterranean diet using a specific questionnaire [19], and physical activity according to the International Physical Activity Questionnaire (IPAQ) [20]. Participants were considered drinkers if they drank more than two standard drink units (SDU) in a day or >14 SDU/week. An SDU is defined as one beer or a glass of wine. Other drinks such as whisky were considered 2 SDU [21]. Metabolic syndrome was analyzed according to the Adult Treatment Panel-III-NCEP definition [22].

After an overnight fast, blood samples were obtained and basic hematological and biochemistry analyses were performed using Siemens Dimension Vista System Flex analysis system at the clinical laboratory (University Hospital Virgen de la Victoria, Málaga, Spain). A sample was immediately frozen and sent for subsequent analysis of high-sensitivity C-reactive protein (hsCRP), insulin, apolipoprotein B (ApoB), and apolipoprotein B48 (Mindray auto analyzer using commercial kits) at Centro de Investigaciones Médico-Sanitarias (CIMES), University of Málaga. Insulin resistance was calculated using the homeostasis model assessment of insulin resistance (HOMA-IR) [23].

### 2.2. Subclinical Arteriosclerosis

All patients underwent a carotid and femoral Doppler ultrasound (Sonoline Antares 10 mH, Siemens, Mountain View, USA). The ecographic segments studied were from the distal segment of the common carotid artery to the proximal segment of the internal and external carotid artery and along both common femoral arteries to their bifurcation. We defined plaque according to the *Progression of Early Subclinical Atherosclerosis* (PESA) study criteria: endoluminal protrusion >0.5 mm, intima-media thickness (IMT) >50% with respect to the proximal zone, and/or IMT 1.5 mm between adventitia and endoluminal light (Figure 1) [24].

The abdominal aorta calcium score (AACS) was assessed using low-dose, contrast-free computed tomography (CT; Philips Brilliance CT 64-slice, Phillips Medical System, Cleveland, Ohio, USA) (Figure 2); calcium was measured using the Agatston score [25]. Abnormal calcium score was categorized as mild (1–100 Hounsfield units [HU]), moderate (101–400 HU), or severe (>400 HU) [26,27]. We divided the abdominal aorta into three areas (Zone 1: from 1 cm above the celiac trunk to 1 cm below the renal arteries; Zone 2: from 1 cm below the renal arteries to 1 cm above the aortoiliac bifurcation; Zone 3: from 1 cm above the aortic bifurcation to 1 cm below it). Patients and controls with plaque detected upon ultrasound examination or an abnormal AACS were categorized as having subclinical arteriosclerosis (SA).

### 2.3. Liver Steatosis

CT parameters used to evaluate fatty liver include the absolute attenuation value (HU_Liver_) and the attenuation value difference between the liver and spleen (HU_L-S_). An HU_liver_ of 48 and a CT_L-S_ of -2 are the threshold values for a 100% specific diagnosis of moderate-to-severe hepatic steatosis (Figure 3) [28,29].

### 2.4. Statistical Analysis

Sample size was calculated for a power of 0.8 and a significance level of 0.05, assuming that the null hypothesis was 0.57 mm in intimo-medial thickness (IMT) in the control group, and the estimated mean was 0.63 mm in the GCT group with a standard deviation of the population is 0.15, representing 50 subjects by group. Data were analyzed using SPSS 22.0 software (IBM, USA). Data are shown as mean ± SD, median (interquartile range), or number (%). All parameters were tested for normality. To compare between groups, Student’s t test or Mann–Whitney were applied. We used the χ^2^ test to compare frequencies between groups. A p value <0.05 was considered statistically significant. To assess which factors were independently associated with SA, a forward stepwise (Wald) binary regression analysis was conducted, taking SA as the dependent variable and all other analyzed factors as covariates.

### 2.5. Ethics

This study was approved by the Ethics Committee of the Hospital Virgen de la Victoria, Málaga on the 22 of june, 2016. All participants provided written informed consent.

## 3. Results

### Study Participants

The study included 103 participants, mean age 48.5 ± 7.2 years (50 GCT survivors, mean age 49.5 ± 7.6 years; 53 age-matched healthy controls, mean age 47.5 ± 6.7 years) from March 2017 to July 2018. Fifty cases of germ cell tumor (GCT) and 53 controls were included. Half of the patients had a seminomatous tumor and 78% were treated with platinum-based chemotherapy. All patients were Caucasian males. Patients were included 14 ± 6.8 years after diagnosis.

Patient demographics and general clinical data results are summarized in Table 1. Both groups had similar prevalences of hypertension, diabetes, and metabolic syndrome, as well as cardiovascular risk assessed by the SCORE system. There was a tendency in the GCT group to have more smokers and past smokers and lower adherence to the Mediterranean diet compared with controls. However, in terms of physical activity, IPAQ scores were higher in the control group than in the GCT group (*p* < 0.05). Patients in the GCT group had higher body weight and waist circumference than controls (*p* < 0.05 and *p* = 0.06, respectively). Concerning the prevalence of metabolic syndrome, the proportions of participants with three or more factors were similar.

Table 2 depicts main analytical data. Patients in the GCT group had higher hemoglobin, platelets, and gamma-glutamyltranspeptidase (GGT), and lower high-density lipoprotein (HDL) cholesterol than controls (*p* < 0.05).

Table 3 shows ultrasound and CT findings. Averaged IMT was nearly identical in both groups; however, there was a trend toward a higher plaque prevalence in the GCT group compared with controls, reaching statistical significance in the right femoral artery (*p* < 0.05). If we consider all the plaques found in ultrasonography, 32 (15%) plaques were identified in the 212 arterial territories investigated in the control group in contrast to 57 (28%) in the 200 territories explored in the GCT group (*p* < 0.05). The GCT group had higher calcium scores than controls only at Area 1 (the upper abdominal aorta; *p* < 0.05). In other areas, total calcium deposition and the prevalence of abnormal AACS were similar between groups. Liver steatosis was significantly higher in the GCT group than in controls (*p* < 0.05).

Only age, ApoB, HDL cholesterol, smoking, and hypertension were associated with SA; being a survivor of GCT and exposure to chemotherapy or radiotherapy were not (Table 4).

## 4. Discussion

The main finding of our study was the observation that after a mean follow-up of 14 years. GCT survivors showed a clear trend toward greater subclinical arteriosclerosis than age-matched controls, reaching statistical significance at the total number of plaques found in the arterial territories explored by ultrasonography as well as the proximal area of the abdominal aorta (AACS). These findings can be explained at least partially by the observation that known vascular risk factors (smoking, hypertension, ApoB, and low HDL cholesterol, all associated with SA in the multivariate analysis) and data related to lifestyle (adherence to Mediterranean diet, physical exercise) were more unfavorable in GCT survivors than in controls.

However, everyone in the GCT group received chemotherapy treatment, and it is well known that cardiovascular disease (and a second neoplasm) are more frequent in GCT survivors in the long term. A higher risk for CVD has been associated with chemotherapy (carboplatin or cisplatin plus etoposide plus bleomycin) and radiotherapy used in the treatment of the disease [6,7,8]. For this reason, we cannot exclude any role of the drugs in the arteriosclerosis of patients. Nevertheless, we found many other contributors. In fact, controls had a stricter adherence to a Mediterranean diet and also performed more physical exercise than GCT survivors. Moreover, the GCT group showed lower levels of HDL cholesterol and higher prevalence of abdominal obesity (two of five components of metabolic syndrome) than controls; all of the components are powerful contributors to arteriosclerosis. The prevalence of metabolic syndrome has been shown to be increased in patients cured of testicular cancer [30], being associated with the use of cisplatin and hypogonadism [31,32]. Despite this, the prevalence of metabolic syndrome was similar between groups.

The serum GGT elevation with normal levels of alkaline phosphatase found in the GCT group is not rare in fatty liver disease [33]; in fact, CT examination revealed that fatty liver was significantly more prevalent among GCT survivors than among controls. Because drinking was not more prevalent among GCT survivors, we may speculate that GCT survivors have non-alcoholic fatty liver disease, which is usually linked to abdominal obesity and insulin resistance syndrome [34]. In fact, the HOMA-IR was higher in GCT survivors compared with controls (*p* = 0.08).

After a careful review of the literature, we did not find any association between chemotherapy or radiotherapy and the risk of non-alcoholic fatty liver disease; thus, we may suggest that this condition is linked to less physical exercise and more abdominal obesity in GCT survivors than in controls. Non-alcoholic fatty liver disease has been associated with a higher risk for SA [35] and cardiovascular events in the general population [36,37].

Data from binary regression analyses did not find any association between cases and SA; however, SA was associated independently with major CV risk factors such as age, smoking, hypertension, ApoB, and HDL cholesterol, as expected. Fasting and postprandial apolipoprotein B48, the main protein in chylomicrons, and its remnants [38] have been associated with SA in many studies in populations both with diabetes [39,40] and without [41,42]. However, apolipoprotein B48 was not associated with SA in our study, even when we removed ApoB from the equation. This lack of association may be linked to the absence of GCT survivors with hypertriglyceridemia. 

Our study was limited by its sample size and also by the fact that a common disease (arteriosclerosis) may be modified by other non-measured factors present in others with lower prevalence as GCT. Thus, future studies with larger numbers of patients and controls should be performed to confirm these results. As strengths, we consider the fact that all of the radiological examinations (carotid and femoral echo Doppler, ACS measurements) were made by the same radiologist, unaware of the clinical condition of participants. Similarly, the clinical data of all included cases were collected by a single professional. Furthermore, nearly 80% of GCT survivors received a platinum-based therapy.

In summary, we found differences with respect to SA between survivors of GCT and healthy people, particularly more plaques in the arterial territories investigated and a greater calcium score in the proximal zone of the abdominal aorta. These differences might be related to metabolic and lifestyle-related factors; however, we cannot rule out the long-term adverse effect of chemotherapy on these patients.

## Figures and Tables

**Figure 1 jcm-09-00971-f001:**
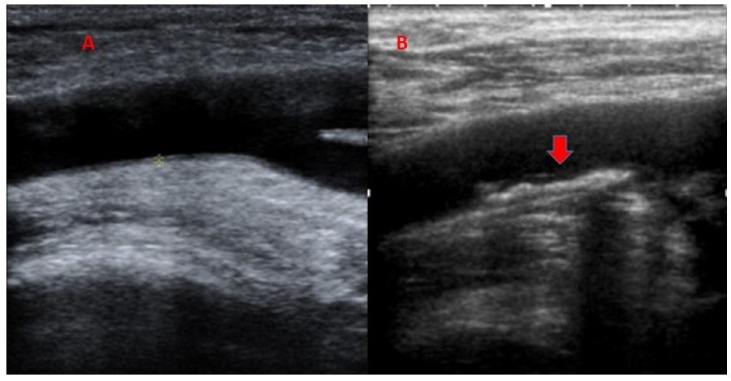
(**A**) Femoral artery of a control subject. (**B**) Femoral artery of a survivor of Germ Cell Tumor (GCT) with an arteriosclerotic plaque.

**Figure 2 jcm-09-00971-f002:**
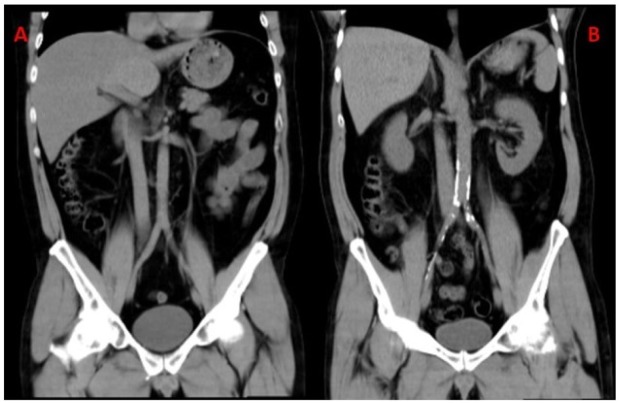
(**A**) Coronal maximal intensity projection (MIP) reconstruction (7 mm thickness) of the abdominal aorta with no arteriosclerosis. (**B**) Abdominal aorta coronal MIP reconstruction with calcium deposition.

**Figure 3 jcm-09-00971-f003:**
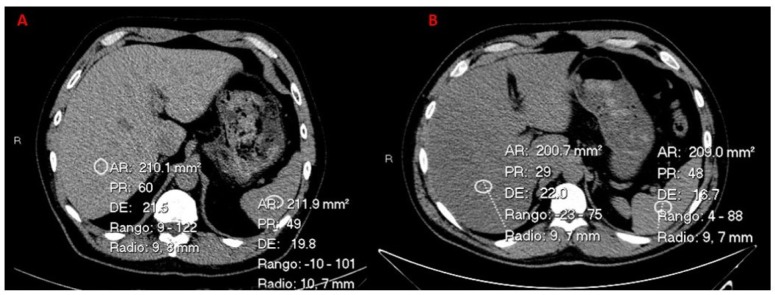
(**A**) Control subject without liver steatosis. (**B**) GCT survivor with liver steatosis.

**Table 1 jcm-09-00971-t001:** General clinical data.

	Total (103)	Controls (53)	Patients (50)	*p* value
Age (Years)	48.5 (7.2)	49.5 (7.6)	47.5 (6.7)	
Age (Diagnosis)			33.2 (8.5)	
Years Since Diagnosis			14.3 (6.8)	
Histology (%)				
Seminomatous			24 (48)	
Non-Seminomatous			26 (52)	
Therapy (%)				
Carboplatin			9 (18)	
Etoposide + Platin			8 (16)	
Bleomicin + Etoposide + Platin			22 (44)	
Others			9 (18)	
Unknown			2 (4)	
Comorbidities (%)				
Smokers	18 (17.5)	8 (15.1)	10 (20)	
Metabolic Syndrome				
Former smokers	23 (22.3)	10 (18.9)	13 (26)	
Drinkers	8 (7.8)	5 (9.4)	3 (6)	
High Blood Pressure	19 (18.4)	9 (17)	10 (20)	
Diabetes	5 (4.9)	2 (3.8)	3 (6)	
Hyperlipidemia	22 (21.4)	13 (24.5)	9 (18)	
Metabolic Syndrome	28 (27)	15 (27)	13 (26)	
Cardiovascular Risk				
Score	1 (0–2)	1 (0–2)	1 (0–2)	
Score ≥ 5 (%)	10 (9.7)	6 (11.32)	4 (8)	
High Cardiovascular Risk (%) *	12 (11.6)	6 (11.3)	6 (12)	
Familiar History of coronary disease	20 (19.4)	14 (26.4)	6 (12)	= 0.06
MEDAS	9.17 (2.5)	9.45 (2.63)	8.86 (2.33)	
Adherence to Mediterranean Diet (%)	76 (74)	43 (81)	33 (66)	
IPAQ (%)				<0.05
Sedentary	15 (14.6)	4 (7.5)	11 (22)	
Medium Physical Activity	44 (42.7)	23 (43.4)	21 (42)	
High Physical Activity	44 (42.7)	26 (49.1)	18 (36)	
Body Weight (kg)	83.5 (11.16)	81.4 (9.86)	85.7 (12.11)	<0.05
Body Mass Index (kg/m^2^)	27.6 (8.48)	26.85 (3.17)	26.91 (3.39)	
Waist Circumference (cm)	100 (12)	98 (9)	103 (15)	= 0.06
Systolic Blood Pressure (mm Hg)	134 (13)	133 (11)	135 (15)	
Diastolic Blood Pressure (mm Hg)	86 (9)	85 (8)	88 (10)	

Data are shown as mean (SD) or n (%) *: score> 5%, diabetics and cardiovascular disease antecedent. MEDAS: Mediterranean Diet Score. IPAQ: International Physical Activity Questionnaire.

**Table 2 jcm-09-00971-t002:** Analytical data.

	Total (103)	Controls (53)	Patients (50)	*p* value
Hb (g/dL)	15.28 (1.8)	14.93 (2.17)	15.66 (1.2)	< 0.05
MCV (fL)	89.91 (6.94)	90.73 (6.4)	89.04 (7.45)	
MCH (pg)	29.85 (2.44)	29.58 (2.4)	30.13 (2.46)	
RDW (%)	13.3 (1.03)	13.6 (0.77)	12.9 (1.16)	
Platelets (10 × 9/L)	239,669 (62,304)	227,792 (65,569)	252,260 (6600)	< 0.05
Total Leukocytes (10 x 9/L)	6550 (1788	6611 (1790)	6527 (1803)	
Neutrophyles (10 x 9/L)	3722 (1490)	3668 (1511)	3781 (1481)	
Glucose (mg/dL)	94 (9)	93 (7)	94 (11)	
Insulin (µmol/L)	9.13 (5.88)	8.19 (4.7)	10.93 (6.83)	
Total Cholesterol (mg/dL)	195 (33)	194 (33)	196 (34)	
HDL Cholesterol (mg/dL)	53 (14)	58 (14)	48 (12)	< 0.05
LDL Cholesterol (mg/dL)	117 (29)	114 (28)	121 (29)	
Triglycerides (mg/dL)	123 (73)	115 (62	132 (83)	
Uric Acid (mg/dL)	5.7 (1.52)	5.79 (1.22)	5.78 (1.8)	
Creatinine (mg/dL)	0.9 (0.15)	0.86 (0.11)	0.94 (0.17)	
eGFR (ml/min/1,73 m^2^)	87 (6)	89 (3)	85 (8)	
Urea (mg/dL)	35 (10)	34 (8)	37 (11)	
Total Bilirubin (mg/dL)	0.7 (0.3)	0.75 (0.33)	0.64 (0.3)	
AST (U/L)	28 (9)	27 (7)	33 (14)	
ALT (U/L)	33 (16)	30 (15)	36 (17)	
GGT (U/L)	26 (16–42.25)	19 (14–30)	38 (21–57)	<0.05
Iron (µg/dL)	97 (36)	94 (35)	99 (37)	
hsCRP (mg/L)	1.92 (0.71–3.17)	1.99 (0.61–2.75)	1.88 (0.91–3.55)	
APOB (mg/dL)	103 (15)	104 (16)	103 (14)	
APOB48 (mg/dL)	7.71 (3.79)	7.50 (2.65)	7.92 (4.71)	
HOMA-IR	2.14 (1.48)	1.90 (1.16)	2.40 (1.73)	= 0.08

Data shown as mean (SD) except GGT and hsCRP shown as median (interquartile range) SI conversion factors: to convert glucose to mmol/l multiply by 0.0555, cholesterol by 0.02586, triglycerides by 0.01129, and urea by 0.357. To convert creatinine to µmol/l multiply by 88.4, uric acid by 59.48, bilirubin by 17.1, and iron by 0.179. To convert C-reactive protein to nmol/l, multiply by 9.524; to convert insulin to mmol/l multiply by 7.175.

**Table 3 jcm-09-00971-t003:** Ultrasound and abdominal computed tomography (CT) data.

	Total (103)	Controls (53)	Patients (50)	*p* value
Ultrasound				
RCCA				
IMT (mm)	0.78 (0.17)	0.80 (0.19)	0.76 (0.15)	
Plaque (%)	24 (16)	10 (19)	14 (28)	
LCCA				
IMT (mm)	0.79 (0.17)	0.80 (0.17)	0.78 (0.17)	
Plaque (%)	18 (12)	8 (15)	10 (20)	
RFA				
IMT (mm)	0.82 (0.35)	0.80 (0.36)	0.84 (0.34)	
Plaque (%)	27 (18)	9 (17)	18 (36)	<0.05
LFA				
IMT (mm)	0.84 (0.34)	0.85 (0.35)	0.83 (0.32)	
Plaque (%)	89 (20)	15 (28)	15 (30)	
Total Plaques	32 (21)	32 (15)	57 (28)	<0.05
Averaged IMT	0.81 (0.20)	0.81 (0.20)	0.80 (0.19)	
Abdominal CT				
Zone 1 (Ag)^1^	0 (0–0)	0 (0–0)	0 (0–1.37)	<0.05
Zone 2 (Ag)	0 (0–51.2)	0 (0–32.5)	0 (0–117.45)	
Zone 3 (Ag)	0 (0–142.3)	0 (0–71.6)	0 (0–277.85)	
Total (Ag)	0 (0–296.6)	0 (0–171)	0 (0–455.07)	
Abnormal AACS (%)	48 (46.61)	23 (43.4)	25 (50)	
Liver steatosis^1^	20 (19.4)	6 (11.3)	14 (28)	<0.05

Data shown as mean (SD), median (IQR), and n (%). RCA: right common carotid artery. LCCA: left common carotid artery. RFA: right femoral artery. LFA: left femoral artery. IMT: intimo-medial thickness. AACS: abdominal aorta calcium score.

**Table 4 jcm-09-00971-t004:** Binary lineal regression analyses taking subclinical arteriosclerosis as dependent variable.

Factor	OR (95% Confidence Interval)
Age (by Year)	1.154 (1.053–1.264)
HDL Cholesterol (by mmol/L)	0.929 (0.887–0.973)
Apolipoprotein B (by mg/dL)	1.068 (1.039–1.146)
Smoking	13.290 (2.370–74.512)
Hypertension	16.00 (1.840–138.402)

Variables not included in the equation: LDL cholesterol, apolipoprotein B48, HOMA-IR, BMI, diabetes, drinking, MEDAS and IPAQ questionnaires, liver steatosis, group (control vs. testicular cancer survivor), and familiar cardiovascular disease.

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
