# Peer review of "Subclinical Arteriosclerosis is Associated with Common Vascular Risk Factors in Long-Term Survivors of Testicular Cancer"

_jcm, 2020, doi:10.3390/jcm9040971_

Round 1

Reviewer 1 Report

Dear author(s), your manuscript covers an interesting topic for CVD risk assessment in survivors of testicular cancer. However there are some major points I would like you to comment on: - First of all your conclusion that GCT survivors show a clear trend toward greater subclinical arteriosclerosis is solely driven by more plaques in the RFA and a higher plaque load in one zone of the abdominal aorta. But in all other vascular territories there was no difference but not even a trend which could be observed. Also in binary regression there was no association found. So your conclusion that atherosclerosis is associated with GCT is not supported by your data and your main conclusion must be questioned. - it seems that the effect size for the association of subclinical atherosclerosis and GCT is small or there is no sig. effect. How was you sample size calculated? For a small effect, 50 vs. 50 patients seem to be underpowered to detect sig. differences. - Patient recruitment is not clearly outlined. Please describe this process in a more detalied from and add additionally a flow chart of study participants. Is the design of the study retrospective or prospective? - Details on statistical analysis are insufficient. Did you test for a normal distribution? - Presentation of the graphs are lacking. In some cases braces are missing. Additionally, it is very difficult to detect the variables which show a stat. difference. Therefore, design of the tables should be improved.

Author Response

Dear author(s), your manuscript covers an interesting topic for CVD risk assessment in survivors of testicular cancer. However there are some major points I would like you to comment on: - First of all your conclusion that GCT survivors show a clear trend toward greater subclinical arteriosclerosis is solely driven by more plaques in the RFA and a higher plaque load in one zone of the abdominal aorta. But in all other vascular territories there was no difference but not even a trend which could be observed. Also in binary regression there was no association found. So your conclusion that atherosclerosis is associated with GCT is not supported by your data and your main conclusion must be questioned.

We agree with the referee that our data comparing both sides only show small differences. To reinforce our conclusion we have reanalysed the data, taking into account that new clinical guidelines confers clinical value to plaques and not to IMT  (Mach et al., 2019). In that sense, we found 32 (15%) plaques among the 212 territories explored in the control group (53 patients by 4 arterial territories) but 57 (28%) plaques among 200 territories in GCT groups, p = 0.009. We have added a row in the Table 3 showing this data and a short comment in results and discussion.   

it seems that the effect size for the association of subclinical atherosclerosis and GCT is small or there is no sig. effect. How was you sample size calculated?. For a small effect, 50 vs. 50 patients seem to be underpowered to detect sig. differences.

We calculated the sample size in advance to initiate the study. According to population studies performed in Spain, the intimo-medial thickness (IMT) in men in 40-49 years-old is 0.57 ± 0.09  mm; and those in 50-59 years-old is 0.63 ± 0.15 mm. We estimated that survivors of GCT had more subclinical arteriosclerosis in terms of IMT and they may have IMT corresponding to a decade later. Thus, to get a power of 0.8, a significance level of 0.05 and assuming that the null hypothesis is 0.57, and the estimated mean is 0.63 with a standard deviation of the population is 0.15, we will need to include 51 subjects in the study. We have added a comment on how sample size was calculated in the statistical paragraph. -

Patient recruitment is not clearly outlined. Please describe this process in a more detailed from and add additionally a flow chart of study participants.  

We have added this sentence in the text: “Among 126 subjects regularly seen in our outpatient clinic, 98 could be contacted by phone and 61 accepted initially. Finally, only 50 patients signed the informed consent and completed the study”.

Is the design of the study retrospective or prospective?

The study was retrospective-

Details on statistical analysis are insufficient. Did you test for a normal distribution?

Yes, we did it. All parameters analysed were tested for normality. In fact, calcium parameters obtained from CT did not follow normal distribution and were expressed as median (IQR).  -

Presentation of the graphs are lacking.

Deeply sorry for the mistake. We have added the graphs in the new version of the paper.

In some cases braces are missing

We have modified braces to make them more visible.

Additionally, it is very difficult to detect the variables which show a stat. difference. Therefore, design of the tables should be improved.

We have changed the design of the Tables to make them more understandable.

Reviewer 2 Report

THIS RETROSPECTIVE OBSERVATIONAL STUDY IS CREATING A HYPOTHESIS:

Risk factors are the same as for cardiovascular diseases, which the authors obserned.

The separation between a disease which has generally high prevalence and a diease with low prevalence is always difficult.

This aspect should be mentioned and shown as a limitation.

Author Response

THIS RETROSPECTIVE OBSERVATIONAL STUDY IS CREATING A HYPOTHESIS:

Risk factors are the same as for cardiovascular diseases, which the authors obserned.

We absolutely agree with the reviewer. Arteriosclerosis is associated with major and classical risks factors (smoking, hypercholesterolemia, … and so on). Some particular chronic diseases (SLE, AIDS, RA among other) increase the risk of an event due to high inflammatory background. In the case of GCT subjects, epidemiological studies clearly showed an increase risk for cardiovascular diseases; the investigators related this to the exposure of the endothelium to platinum-based chemotherapy. We can not rule out that association but is clearly showed in our study that main differences in subclinical arteriosclerosis are associated with classical risk factors and, very importantly, with life-style.

The separation between a disease which has generally high prevalence and a diease with low prevalence is always difficult.

This aspect should be mentioned and shown as a limitation.

Absolutely, we include a sentence in the paragraph devoted to limitations.

Reviewer 3 Report

The present manuscript contains detailled analyses of 50 GCT patients and 53 control subjects. The strengt of the paper is detailed analyses of blood vessels utilizing several parameters. Maybe a larger cohort would have been useful to provide more valid data. Also testosterone levels would be valuable parameter at this setting and explain waist circumference, metabolic syndrome...

Only some minor notes to the manuscript that should be addressed.

Page 1 29-31 Authors state …, but not with exposure to chemotherapy or radiotherapy. Next inclusion criteria page 2 lines 59-60 GCT treated with chemotherapy? Some discrepancy?

Page 2 line 47 plaques break and…. I think it would be more accurate to state plaques break or…

Symptoms may occur due to either plaque rupture, occlusion or significan stenosis of the vessel or thrombosis ?

Table legens have been mixed? Table 1 includes on my versio legend for table 2, table 2 legend for table 3. Legends for table 1 and 4 missing?

Figures 1-3 were missing on my versio. It would be nice to review those also.

Author Response

Reviewer 3

The present manuscript contains detailled analyses of 50 GCT patients and 53 control subjects. The strengt of the paper is detailed analyses of blood vessels utilizing several parameters. Maybe a larger cohort would have been useful to provide more valid data. Also testosterone levels would be valuable parameter at this setting and explain waist circumference, metabolic syndrome...

We really appreciated the comment of the referee; for sure, a larger cohort would increase the validity of our data, but in overall they are consistent. We also agree that testosterone levels could explain part of our results; unfortunately, we are not now in conditions to measure the hormone.

Only some minor notes to the manuscript that should be addressed.

Page 1 29-31 Authors state …, but not with exposure to chemotherapy or radiotherapy. Next inclusion criteria page 2 lines 59-60 GCT treated with chemotherapy? Some discrepancy?

We agree with the reviewer; in order to avoid confusion, we only mention chemotherapy in the abstract, which was in fact the inclusion criteria.

Page 2 line 47 plaques break and…. I think it would be more accurate to state plaques break or…

Symptoms may occur due to either plaque rupture, occlusion or significan stenosis of the vessel or thrombosis?

We agree with the referee that occlusion by a significant stenosis (i.e. stable angina) or thrombosis over a ruptured plaque (acute coronary syndromes) are needed to trigger symptoms. We have modified the sentence to be more accurate.

Table legens have been mixed? Table 1 includes on my versio legend for table 2, table 2 legend for table 3. Legends for table 1 and 4 missing?

We have reviewed the Tables legends and are now correct.

Figures 1-3 were missing on my versio. It would be nice to review those also.

We are deeply sorry for the mistake. Figures are now in the revised manuscript.

Round 2

Reviewer 1 Report

Dear author(s),

I am satisfied with the changes in terms of the description of the results and statistical section. However, the design of the tables (sometimes braces missing, somtimes there is a comma instead of a point- see for example MEDAS in table 1) should be improved.

Author Response

I am satisfied with the changes in terms of the description of the results and statistical section. However, the design of the tables (sometimes braces missing, somtimes there is a comma instead of a point- see for example MEDAS in table 1) should be improved.

Thanks for highlight the mistakes. We have made changes in the Tables, having replaced commas by points, added braces missing  and also a third column with the p value.